# Biological Bases of Beauty Revisited: The Effect of Symmetry, Averageness, and Sexual Dimorphism on Female Facial Attractiveness

**Alex L. Jones** [1,*] and **Bastian Jaeger** [2]

1   Department of Psychology, Swansea University, Singleton Park, SA2 8PP Swansea, South Wales, UK
2   Department of Social Psychology, Tilburg University, 5037 AB Tilburg, The Netherlands; B.Jaeger@uvt.nl
*   Correspondence: alexjonesphd@gmail.com

**Abstract:** The factors influencing human female facial attractiveness—symmetry, averageness, and sexual dimorphism—have been extensively studied. However, recent studies, using improved methodologies, have called into question their evolutionary utility and links with life history. The current studies use a range of approaches to quantify how important these factors actually are in perceiving attractiveness, through the use of novel statistical analyses and by addressing methodological weaknesses in the literature. Study One examines how manipulations of symmetry, averageness, femininity, and masculinity affect attractiveness using a two-alternative forced choice task, finding that increased masculinity and also femininity decrease attractiveness, compared to unmanipulated faces. Symmetry and averageness yielded a small and large effect, respectively. Study Two utilises a naturalistic ratings paradigm, finding similar effects of averageness and masculinity as Study One but no effects of symmetry and femininity on attractiveness. Study Three applies geometric face measurements of the factors and a random forest machine learning algorithm to predict perceived attractiveness, finding that shape averageness, dimorphism, and skin texture symmetry are useful features capable of relatively accurate predictions, while shape symmetry is uninformative. However, the factors do not explain as much variance in attractiveness as the literature suggests. The implications for future research on attractiveness are discussed.

**Keywords:** faces; attractiveness; symmetry; machine learning; averageness; dimorphism

## 1. Introduction

The qualities that make an individual attractive have been an area of scholarly inquiry for thousands of years, across many societies and disciplines [1]. Classical theories of attractiveness included the belief that "beauty is in the eye of the beholder" [2], an idea that Darwin himself supported. However, in recent decades, one of the greatest successes in understanding the qualities of attractiveness have been in the area of psychology, through the application of evolutionary principles to inform hypothesis generation and interpretation [3,4]. This approach has amassed significant evidence that there are specific factors that observers seem to find particularly attractive, especially in faces. While individual variation does influence perceptions of beauty [5], these factors seem preferred cross culturally [6,7], indicating a biological basis for attractiveness.

Research has identified symmetry, averageness, and sexual dimorphism as driving factors of facial attractiveness, though the extent to which each is preferred varies over different contexts [3]. The influence of each trait on facial attractiveness has been informed by biology [8,9], and each trait has been linked to various markers of genetic fitness [10], reproductive success [11], and positive life history outcomes [12]. These findings have been interpreted in the context of biological signalling

theories. For example, symmetry and averageness are honest signals, affirming the health of an individual, while sexual dimorphisms, particularly for males, are handicaps grown under stressed immunity. The link to psychological perceptions of attractiveness is clear: selecting a mate with these qualities improves the health of offspring by avoiding partners who may have a high parasite load or low genetic heterozygosity [8]. However, these findings have recently been called into question, with numerous putative links between these traits and relevant life outcomes failing to replicate in larger samples [13–15]. The current evidence for each is reviewed below.

### 1.1. Symmetry

Human faces are bilateral structures and show high degrees of symmetry. However, as in many other organisms, human faces exhibit various amounts of fluctuating asymmetry (FA), or deviations from perfect bilateral symmetry. Individuals with a greater FA are thought to have greater development instability due to environmental insults in the form of pathogens, toxins, or genetic mutations or can indicate a lack of genetic defence overall [16,17]. There are several convincing lines of evidence supporting this biologically-derived prediction. For example, school children from poorer districts have greater facial asymmetries than those in wealthier areas, supporting the notion that a harsher environment produces more developmental insults [18]. Research has supported that humans are sensitive to deviations in symmetry, perceiving more asymmetrical faces as less attractive [19–22]. However, more recently, large-scale studies have found no associations between facial symmetry and evolutionary relevant markers [23]. For example, measured facial asymmetry shows no association with self-reported health [15], and perceived facial symmetry shows no relationship with oxidative stress or immune function [14]. In addition, the magnitude of the effect of symmetry is relatively small, with meta-analytic estimates concluding that it contributes the least amount of variance to facial attractiveness, with a mean effect size $r$ of 0.25 [4].

### 1.2. Averageness

In the same way that symmetry is a plausible cue to traits like parasite load or mutations, facial averageness is related to more diverse genetics, or heterozygosity, which in turn improves immune function [24,25]. Thus, the biological prediction is that faces closer to the average configuration should have stronger immune function. This notion has received some support in the literature, with adult facial averageness being related to childhood health [26] and current health [15], while chromosomal disorders are related to deviations from average facial proportions [27]. Faces closer to the average are perceived as more attractive, consistent with the prediction of honest signalling [26,28,29]. However, some evidence suggests that this is due to faces with significant deviations from average being rated as far less attractive, rather than faces with greater averageness being rated as more attractive—that is, distinctiveness is unattractive rather than averageness itself being attractive [30]. However, much like symmetry, recent findings have called into question the relationship between averageness and health, with no relationship emerging between facial averageness and oxidative stress, immune function [14], or salivary immunoglobulin [13]. In spite of this, facial averageness is heritable; a vital element of any evolutionary argument [31]. Thus, averageness is a significant predictor of facial attractiveness [4], though its role as an honest signal of health or genetic quality is less clear.

### 1.3. Sexual Dimorphism

While average faces seem attractive, the most attractive faces are not average [32]. Rather, there are specific deviations from the average configuration that lead to increased perceptions of facial attractiveness in the direction of sex-typical appearances [9]. While these morphological patterns are referred to widely in the face perception literature as femininity and masculinity, a more biological interpretation is of a degree of maleness and femaleness, with dimorphism being a state that is present or absent [33]. To draw comparisons with the wider literature, the terms "femininity" and "masculinity" are used throughout. Increased masculinity and femininity in male and female faces respectively

leads to consistent and significant increases in perceived attractiveness [4]. Sex differences in appearance may also relate to individual differences in health and reproductive potential. For example, the immunocompetence hypothesis suggests that male facial masculinity is a handicap signal—only the healthiest males can afford to "spend" testosterone on these ornaments while suffering immune suppression under high testosterone loads, a theory borrowed from ethology [34]. Similarly, facial femininity may be a cue to fertility, given the relationships between feminine characteristics and circulating oestrogen [35]. There is also evidence that females with more feminine faces are actually healthier [36]. However, as with both symmetry and averageness, recent evidence has questioned the honest signalling or handicap hypotheses of sexually dimorphic facial features. For example, testosterone does not seem to suppress immune function in human males [37], and masculinity does not seem particularly attractive to females [38,39]. Facial femininity shows no relationship with actual health [15] or immune function [13], and more attractive women do not have higher levels of progesterone or estradiol [40].

## 1.4. Methods in Attractiveness Research

While facial attractiveness seems reliably predicted by symmetry, averageness, and sexual dimorphism, the evidence that these factors reflect evolutionarily relevant signals is currently weak, with many established findings failing to replicate with larger sample sizes and robust methods [41]. Nonetheless, regardless of their evolutionary utility, the factors do seem to alter perceptions of attractiveness. However, it is worth considering the methods used in attractiveness research to properly understand the strength of the evidence for each factor influencing attractiveness. This is especially important in light of the approaches that emphasise more robust statistical estimates of effect sizes rather than significance [42] and the current replication crisis affecting all areas of psychology [43].

### 1.4.1. Two-Alternative Forced Choice Paradigms

A significant proportion of facial attractiveness research uses a combination of image manipulation and forced choice paradigms. Typically, faces are altered along a particular dimension using shape and texture transforms [44] and presented next to their counterpart, altered in the opposite direction along the same dimension, or next to an unmanipulated version. For example, studies reporting the attractiveness of symmetry in faces have participants select which face they perceived as most attractive between an original, unmanipulated image and a perfectly symmetrical version [20] or demonstrate the effect of averageness through similar methods [45]. Research examining the role of sexual dimorphism relies most often on comparing a feminised and masculinised version of the same face, achieved by altering a face along a vector between average female and male faces [46,47].

This two-alternative forced choice approach is very useful for two reasons. The first is that it is a manipulation method, which is necessary to establish the causality from the correlations observed between the perceived attractiveness and the measurement of these factors [8,28]. The second is that since the methodology holds all traits constant aside from the one of interest, it isolates the effect of interest from the vast dimensions in which faces can vary. Indeed, symmetry, averageness, and sexual dimorphism all show correlations between each other [28,48], though they do contribute independently to perceived attractiveness [49–51]. This combination of sensitivity and ability to establish causation has seen the technique applied in a significant number of studies that make claims about the mechanisms of attractiveness perceptions, from basic properties through to various contextual scenarios [52–63].

However, this methodology has been justifiably criticized for having poor ecological validity [64]. Humans do not make attractiveness judgments by comparing nearly identical versions of a potential mate and selecting the version with the more appealing trait. Rather, attractiveness judgments in the real world are likely a combination of many factors. More broadly, while the sensitive test of the forced choice methodology is capable of producing statistical significance, its ability to explain real world attractiveness judgments and preferences may be limited. Furthermore, in the case of sexual dimorphism, in which the methodology has been most widely applied [4], it is only possible to draw

relative claims, rather than absolute ones, as feminised faces are presented in contrast to masculinised faces. It is, therefore, unclear whether observers actually find more feminine faces more attractive or find masculinised faces less attractive. Research in other areas of face perception has found that when examining a trait of interest using the forced choice methodology, comparing manipulations to unaltered faces results in observers avoiding the low levels of a trait, rather than demonstrating a preference for high levels of a trait [65,66], which is the way the findings are typically interpreted. This pattern is consistent with the overgeneralisation hypothesis [30]: attractiveness perceptions are geared towards avoiding poor choices of mates rather than seeking out the highest quality partners. It is worth considering that a significant proportion of what is known about the role of these factors of attractiveness comes from this sensitive yet artificial method.

### 1.4.2. Ratings Studies

Another approach to studying how the factors alter attractiveness causally is through ratings studies, where observers are presented with a set of faces with manipulations of the factors and asked to rate them on their attractiveness. This approach has provided converging evidence to forced-choice studies, with observers' ratings of attractiveness linearly increasing from decreased averageness and symmetry, through unmanipulated faces, to faces with increased levels of the trait [51]. These rating tasks speak more to how attractiveness may be perceived in the real world, in that faces are considered in isolation and graded along some response scale. Indeed, studies utilising natural, non-manipulated faces across a range of stimulus types provide supporting evidence that more symmetrical and average faces are more attractive [8,22,67–69]. However, as with forced-choice paradigms, the exact method used to collect ratings is important for the interpretation of any effects. It is a typical feature of ratings studies using a manipulation to present all levels of the manipulated variable to participants in a single session [51,70,71]. While this increases statistical power, it also introduces the significant possibility of carryover effects. For example, if a participant views a face with reduced averageness and then, a few trials later, views the same identity with increased averageness, it is unsurprising their ratings increase—again, this methodology produces only a relative interpretation, albeit more indirectly. The number of manipulations and number of faces can compound these carryover effects, with the least desirable outcome involving higher numbers of the former to lower levels of the latter, making any change between identities noticeable [70]. Separating manipulations into two groups, so observers see only one level of a manipulation, alleviates the carryover effects but introduces another issue: that the variance in the manipulation of interest is now reduced as it is present in all faces [72]. More simply, how important is a trait like symmetry to perceived attractiveness if all faces are perfectly symmetrical?

Researchers have used various methods to get around these issues. For example, Morrison and colleagues [73] examined the role of emotional expression in perceiving attractiveness using 30 faces and seven expressions. They presented their stimuli in such a way that each participant rated all face *identities* (i.e., all 30 faces), but the emotional expression of each face they saw was determined at random. They found that the variation attributable to expressions was lower than that attributable to the basic differences in appearances between faces. This kind of approach has also been applied to the effect of cosmetics on attractiveness ratings, finding that, in contrast to previous work that focused on significance [70], the effect sizes of self-applied cosmetics on attractiveness is very small, especially when compared to differences between faces [74], a finding that replicated even when using highly attractive models [75]. That is, even a relatively drastic change in appearance contributes little to attractiveness perceptions. Taken together, the evidence suggests that ratings studies that utilise manipulations to establish causality in attractiveness perceptions may inflate the effects of a given trait by allowing participants to view all the manipulations in a single setting.

A final point is that the majority of manipulation studies do not take into account how faces are influenced by the manipulations themselves. For example, individual faces closer to the average may change little when being manipulated, while those further away might be affected much more strongly. This variation in how manipulations affect faces is typically not modelled, as ratings are averaged over

stimuli or participants before the analysis occurs. Though untested, it is possible that faces with greater changes from the manipulation are driving the effects known in the literature, while the majority of faces may remain unchanged. Indeed, between-face variability is a large source of variance when compared to manipulations [73–75]. Recent efforts have modelled this stimulus variation as a random effect using linear mixed models, finding modest effects of important factors like skin condition on attractiveness [76]. These approaches are capable of quantifying the variance that arises from faces and participants, which is necessary because some participants may give lower or higher average ratings irrespective of the faces they see. Concurrently, some faces may receive lower or higher average ratings regardless of how they are manipulated due to differences in appearances. Random effects allow for a more precise estimate of measures as they can model these components. How symmetry, averageness, and sexual dimorphism manipulations cause variation in ratings studies is unknown, but modelling it successfully would better isolate the effects of the factors themselves. These models can also be extended to forced choice paradigms, where the variation in facial appearance from manipulations can be modelled [77].

*1.5. The Current Studies*

The above evidence provides a critical review of the current literature in facial attractiveness research with regards to the relevance of symmetry, averageness, and sexual dimorphism. With recent evidence calling into question their evolutionary utility, it is a fitting time to examine and carefully quantify their actual role in perceptions of attractiveness. Across three studies, the role of symmetry, averageness, and sexual dimorphism in perceiving female facial attractiveness is examined using refined methodology and modernised analyses, with the aim of understanding the size of the effect, the direction, and the predictive utility of these measures in understanding attractiveness. Female faces are used as they typically show higher agreement among observers, and theories indicate attractiveness and biological function is more closely tied in females than males [4,41]. It is important to note that studies that have set out to quantify the effect sizes of these factors of attractiveness in unmanipulated faces with sophisticated methods have found that they explain very small amounts of variance generally [78,79].

For all experiments reported here, the full data and Python code used to analyse it are available from the Supplementary Materials.

## 2. Study One: Two-Alternative Forced Choice Methods

The main criticisms of two-alternative forced choice methods is that they suffer poor ecological validity [64] and that they provide only relative information. It is not clear whether observers select one face because the other is particularly unappealing rather than finding a given face more attractive. Moreover, there is the possibility that subsets of faces that are greatly affected by the manipulation can drive the effects.

Here, a set of four two-alternative forced choice experiments are carried out. Participants compared unmanipulated faces to more symmetrical, more average, more feminine, and more masculine counterparts. The final two experiments are particularly novel, as they establish the direction of the effect observed in many other experiments, while the former serve as conservative replications of earlier work [20,45], with a strong focus on the size of the effect. Each study is also analysed using a binomial mixed effects model, allowing the capturing of variance in both the responses of participants as well as the way the faces themselves are altered by the manipulations.

In line with existing research, it is predicted that symmetrical faces will be preferred to unmanipulated faces and more average faces will be preferred to unmanipulated versions. In line with meta-analytic estimates [4], the manipulated averageness should show a larger effect than the manipulated symmetry (which results in total bilateral symmetry rather than simply fluctuating asymmetry). It is predicted that feminised faces will be preferred to unmanipulated faces and also that

unmanipulated faces will be preferred to masculinised faces. It is unknown what the effect size will be, since no study has compared the absolute difference between masculinised and feminised faces before.

*2.1. Method*

2.1.1. Participants

Four separate samples of participants were recruited for this study, one for each of the traits. Participants were recruited through social media platforms, and data collection was carried out for the duration of one month, with data collection stopping after this time. Thirty-five participants completed the symmetry task (age $M = 29.06$, $SD = 8.19$; 12 females), a different 30 participants completed the averageness task (age M = 27.70, $SD = 7.72$; 11 females), a further 32 participants completed the femininity task (age $M = 33.06$, $SD = 10.66$; 8 females), and a final 31 participants completed the masculinity task (age $M = 32.58$, $SD = 10.24$; 11 females).

2.1.2. Stimuli

One hundred and nine females (age $M = 20.54$, $SD = 2.26$) posed as models. Each was photographed with a neutral expression, without cosmetics or facial jewellery, using a Canon EOS 5D Mark II camera with a Canon EF 24-105 mm lens (Canon, Ōta, Tokyo, Japan). All the camera settings were kept constant between exposures, including the ISO rating (100), lens aperture (F8.0), focal length (65 mm), and shutter speed (1/100). The models were compensated £6 for their participation. All the faces were manually delineated with a set of 160 points in JPsychomorph [44] to outline the facial appearance.

**Symmetry manipulations**: The faces were manipulated for symmetry following established methods described in the literature, which alter the shape symmetry without changing the texture or skin condition [20,45], using the JPsychomorph package [44]. To manipulate the symmetry, the positions of bilateral points (i.e., the corner of the left eye and corner of the right eye, the corners of mouth, the cheekbones, etc.) were mirrored, and the results were averaged, producing a symmetrical face shape. The original face texture was then mapped to this new symmetrical shape, retaining its identity and texture appearance. This procedure was carried out for each face, generating a symmetrical version for each.

**Averageness manipulations**: A random sample of 73 female faces were taken from the full sample of 109 faces (to match a male average; see below) and were averaged together to create an average female facial shape. Using this face, the linear difference between each face and the average was computed (i.e., a difference measure of how far each face was from the average). To manipulate facial averageness, a 50% scaling of the difference between each individual face and the average was added to the individual face, effectively shifting each face towards the average whilst retaining its unique identity. This procedure was carried out using JPsychomorph [44] and has been used before in the literature [60].

**Femininity and masculinity manipulations**: To manipulate sexual dimorphism, 73 photographs of male faces, taken from the same photoshoot that yielded the female models, were averaged. The linear difference between the female and male averages represents a sexual dimorphism dimension that the individual faces were transformed along, achieved by adding a ±50% scaling of the difference between each average to each individual face shape, which their individual textures were then mapped to. This technique is widely used for sexual dimorphism transforms in the literature [9,80], allowing for faces to subtly change in appearance whilst retaining their identity.

The above steps produced four versions of each face: more symmetrical, more average, more feminine, and more masculine.

### 2.1.3. Procedure

The participants followed a link to an online experiment (testable.com), where they were shown the 109 pairs of faces for the particular experiment (e.g., symmetry, averageness, femininity, or masculinity) in a random order. The side of the screen each face appeared on was counterbalanced between participants. Participants were instructed to "Click on the face you find most attractive." Participants saw only one manipulation, comparing it to the original, unmanipulated face.

### 2.2. Results

There are two analytical approaches taken here. The first is using standard inferential statistics to compare the frequency of selected faces to chance, e.g., a binomial test, and the second uses a binomial mixed effects model to capture the variance in stimuli and participants.

Given there were more faces than participants, the proportion of trials in which the manipulated face (symmetrical, averaged, or dimorphic) was chosen was computed across participants, yielding a proportion for each face. This distribution was then compared to chance, i.e., 50%, using a one-sample $t$-test. With 109 faces, an alpha at 0.05 (two-tailed), and 90% power, this test is sensitive enough to find effects as small as $d = 0.31$. The means are shown in Figure 1.

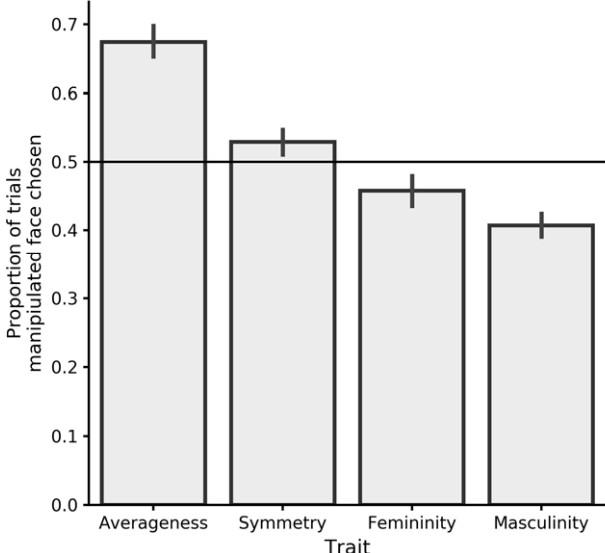

**Figure 1.** The proportion of trials for which the manipulated version of the face was chosen. The horizontal line represents the chance at 50%. The error bars represent the 95% confidence intervals.

The more symmetrical version of a face was selected more often than chance ($M = 0.53$, $t(108) = 3.13$, $p = 0.002$, and $d = 0.30$), and the more average version of the faces was also selected more often than chance ($M = 0.67$, $t(108) = 15.18$, $p < 0.001$, and $d = 1.45$), confirming the hypothesis and replicating previous research [45]. For masculinity, the manipulated version was chosen significantly less than the natural, unmanipulated face ($M = 0.41$, $t(108) = 10.61$, $p < 0.001$, and $d = 1.02$), as predicted. This was also the case for femininity, with the more feminine face being chosen less often than the unmanipulated face ($M = 0.46$, $t(108) = 3.58$, $p = 0.001$, and $d = 0.34$).

However, the above tests do not consider the variance that each participant and stimuli contributes to the overall model. It may be that the effects observed are driven by a subset of faces that are strongly affected by the manipulation, or a subset of participants who find a particular trait attractive. These may be enough to drive a mean difference in the tests. Applying binomial mixed effects models allows these variances to be incorporated when predicting the binary outcome of whether the manipulated face was chosen or not. Given the simple design of the experiments here, all mixed effects models were specified with a constant and with random intercepts for faces and participants.

For symmetry, a significant positive effect emerged ($b$ = 0.12, 95% CI [0.01, 0.22], *SE* = 0.05, $z$ = 2.22, and $p$ = 0.026), indicating symmetrical faces were chosen more often than their unmanipulated versions. For averageness, the mixed model revealed a significant positive effect ($b$ = 0.79 [0.59, 0.99], *SE* = 0.10, $z$ = 7.84, and $p$ < 0.001), again indicating that more average faces were chosen more than the unmanipulated versions. For masculinity, the model fit revealed a significant negative effect ($b$ = −0.38 [−0.51, −0.26], *SE* = 0.07, $z$ = −5.85, and $p$ < 0.001). The increased masculinity led to more choices of the unmanipulated face. Finally, for femininity, the increased femininity led to more choices for the unmanipulated face ($b$ = −0.18 [−0.33, −0.03], *SE* = 0.08, $z$ = −2.29, and $p$ = 0.022). In all models except for masculinity, the inclusion of random intercepts for both participants and faces produced the best fit (all $X^2$ > 5.28 and $p$s < 0.001). This suggests that the symmetry, averageness, and femininity manipulations produced significantly different effects across faces and that participants significantly varied in their responses to these changes. However, for masculinity, the inclusion of random intercepts for faces did not improve the model fit from just random intercepts for participants. This indicates that while participants varied in their responses to the faces, the masculinity manipulation affected all faces similarly.

*2.3. Discussion*

The results from this set of experiments replicate and extend previous research using the two-alternative forced choice paradigm. First, a preference for symmetrical face shapes was apparent. One of the aims of the current study is to quantify the size of factors of attractiveness, and for symmetry, this is not large ($d$ = 0.30). Previous work examining symmetry in this way has found an effect, replicated here, but not focused on the actual size of that effect [20,45,81] which is small. Indeed, given the sensitivity of the forced choice paradigm, this interpretation casts further doubt on the relevance of symmetry in perceiving attractiveness aside from its evolutionary utility. However, this does not rule out symmetry as a useful predictor of attractiveness. For example, several meta-analyses report small effect sizes of symmetry and indicate this is to be expected, given the indirect nature of its measurement of developmental stability as an indirect measure of genetic quality [82–84].

Averageness showed a very large effect ($d$ = 1.45). It is worth considering the magnitude of this effect size in absolute terms, particularly amongst psychological research. Cohen himself described effect sizes of 0.8 as "grossly perceptible", comparing them to the differences in height between 13- and 18-year-old females [85]. Averageness results in a very large increase in attractiveness perceptions using this paradigm and produced the largest effect out of all the factors of attractiveness. However, its absolute relevance cannot be tested using this paradigm.

The sexual dimorphism results indicated that while masculinity is clearly not preferred in female faces ($d$ = 1.00), neither is increased femininity ($d$ = 0.34). As the forced choice paradigm can only establish relative preferences, research teasing apart the direction of manipulation effects have often found that participants exhibit a relative preference for the unmanipulated stimuli to the "decreased" version rather than a preference for the "increased" version [65,66,76]. That is, rather than preferring the increased femininity in faces, participants dislike masculinity. This fits closely with the overgeneralisation hypothesis [30], that suggests that perceptions of attractiveness are a sensitivity to "bad genes" rather than a sensitivity to the best genes. However, the result that the increased femininity was aversive is surprising. One possibility for this is that the amount of femininity introduced by transforms is outside of the range seen in typical faces, making it appear unnatural. Studies involving manipulating sexual dimorphism in faces show a modest preference for increased femininity [86]. Altogether, these results show that claiming preferences for femininity using a two-alternative forced choice task is unwise, given it is likely produced from a relative comparison and cannot speak to how femininity may be valued in a real scenario.

Finally, the use of binomial mixed effects models provided converging evidence with standard statistical approaches—all models indicated similar magnitudes and directions to the effects shown by one-sample *t*-tests. However, the added advantage of this approach is to model the sources of variation from faces, how their differing attractiveness levels are altered by a given manipulation, and how

participants' own attractiveness preferences shape their decisions. All the models indicated that participants respond differently to the manipulations, and all the models except masculinity indicated the manipulations affected the faces to different extents. Researchers employing manipulations should pay attention to the number of stimuli faces, as a small number of stimuli are more liable to produce a significant effect of a manipulation if a subset of the faces change drastically with the manipulation.

## 3. Study Two—Ratings

The findings of Study One point to the limitations of the two-alternative forced choice method. First, they suggest that the absolute effect sizes of some traits may be small, while others are very large. Given the sensitive nature of the forced choice paradigm, this means that the small effects observed here are likely to be even smaller under more realistic conditions and that large effects may be overstated. This is particularly the case for symmetry and averageness. Second, the results indicate that the relative comparisons of the forced choice approach may be misleading. Rather than a strong positive effect for femininity, a strong negative effect for masculinity was observed, with an additional weaker negative effect for femininity.

Ratings studies ameliorate many of these issues by providing a more naturalistic setting. In Study Two, another four studies are conducted, testing the effects of symmetry, averageness, and sexual dimorphism using the randomised presentation approach used by others to isolate specific effects without any carryover effects [73–75]. Rather than have the participants rate all the levels of the manipulation and, thus, risk comparison effects, the participants rate all the identities, but whether they see a manipulated version of the face or the original is determined randomly. Ratings can then be compared between conditions, using conventional statistical approaches as well as linear mixed models to account for the variance in faces and participants. It is worth noting that this is a conservative test in complete contrast to the two-alternative forced choice approach. If the evidence for the factors of attractiveness is robust, then the participants should be sensitive to the increased levels of symmetry, averageness, and dimorphism under these conditions. Given the experiment utilises manipulations, it is also possible to conclude the factors are genuinely affecting attractiveness, compared to correlational studies where other variables may influence a judgment.

For symmetry, it is predicted that there will be an increase in the rated attractiveness, though very small, under these conditions. For averageness, there should be a significant increase in the rated attractiveness with a modest effect size. For masculinity, the ratings of attractiveness should decrease. Existing theories indicate femininity should increase attractiveness ratings, though the results from the prior study call this into question.

### 3.1. Method

#### 3.1.1. Participants

Four separate samples of participants were recruited as in Study One, one for each of the four manipulations that were tested separately. The testing took place over the course of a single semester, and recruitment was stopped at the end. Forty-one participants completed the symmetry ratings experiment (age $M = 19.68$, $SD = 1.31$; 24 females), the averageness experiment (age $M = 19.39$, $SD = 1.02$; 27 females), and the femininity experiment (age $M = 19.39$, $SD = 1.18$; 39 females). Fifty participants completed the masculinity experiment (age $M = 20.94$, $SD = 5.28$; 39 females). The participants completing the symmetry, averageness, and femininity studies were students at a North American liberal arts college who received a course credit for their participation. Participants completing the masculinity study were students at a university in the United Kingdom, also compensated for their participation with a course credit. Recent simulations have shown the number of participants in each condition is sufficient to provide stable average scores [87]. The origins of the samples are unlikely to cause differences in responses to the manipulations, as similar attractiveness results are obtained from experiments conducted in Westernized countries like the UK, Australia, and the United States [4].

### 3.1.2. Stimuli

The experiments here used exactly the same models and stimuli as Study One but were presented differently. The manipulations carried out on the original faces for Study One were utilised here.

### 3.1.3. Procedure

The participants rated the models for attractiveness on desktop computers in a lab-based environment. Images of the models were presented in a random order, and each model was rated only once (109 trials in total), under a randomly selected manipulation condition (unaltered or symmetrical, more average, or more dimorphic, depending on the study). Participants were asked "how attractive is this face?" and indicated their responses using a single mouse click on a scale below the face, which ran from one (not very) to seven (very much so). The stimuli were presented using a custom script in PsychoPy [88,89].

### 3.2. Results

Several analysis approaches were conducted here. First, ratings were averaged across observers to compute a single attractiveness rating for each face under both presentation conditions, i.e., normal or manipulated, allowing for a direct comparison of the attractiveness ratings with a paired samples *t*-test. In the case of null results, a Bayesian inference was used to categorise the strength of the evidence in favour of the null hypothesis. Third, linear mixed models were employed to model the variability associated with participants who would likely show differences in their baseline attractiveness ratings, as well as the variability associated with faces that differ in their baseline attractiveness and, thus, would be affected by the manipulations differently. These approaches allow for more precise characterisations of the effects. The test is powered to detect effects as small as $d = 0.31$ as before, assuming a power of 0.90, an alpha of 0.05 (two-tailed), and a sample of 109 faces. The mean ratings for each face, under each manipulation and trait condition, are shown in Figure 2.

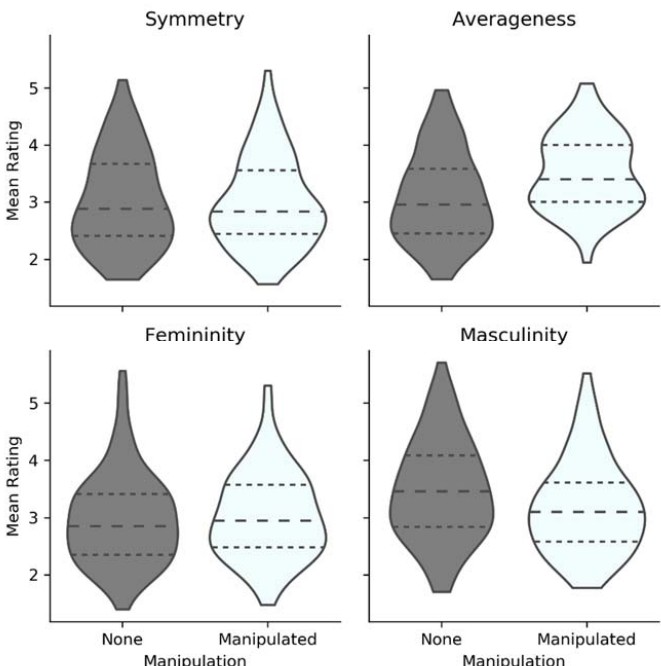

**Figure 2.** Violin plots illustrating the distribution of the mean ratings for each face under each manipulation condition for each trait: The mean is represented with the middle dashed line, and the interquartile ranges are represented by the outer dashed lines.

For symmetry, the faces showed no significant differences between the normal (*M* = 3.02, *SD* = 1.49) or symmetrical versions (*M* = 3.00, *SD* = 1.46); $t(108) = 0.68$, $p = 0.498$, and $d = 0.07$. For averageness, the faces were rated as significantly more attractive when manipulated (*M* = 3.50, *SD* = 1.36) as compared to normal (*M* = 3.10, *SD* = 1.36); $t(108) = 9.14$, $p < 0.001$, and $d = 0.88$. For femininity, the faces showed no significant differences between the normal (*M* = 2.98, *SD* = 1.58) or more feminine versions (*M* = 3.03, *SD* = 1.59); $t(108) = 1.11$, $p = 0.271$, and $d = 0.11$. Finally, for masculinity, the faces were rated as less attractive when masculinised (*M* = 3.15, *SD* = 1.44) compared to normal (*M* = 3.51, *SD* = 1.37); $t(108) = 8.10$, $p < 0.001$, and $d = 0.78$.

The above results indicated no effect for symmetry and femininity against the predictions made but resulted in the clear support for the hypotheses generated for averageness and masculinity. The latter two traits showed the largest effect sizes under a forced choice presentation that survived the more conservative approach here, but the smaller effects of femininity and symmetry did not. However, it may be the case that the effects are too small to detect with the design and sample size used. To offer a more robust conclusion, Bayesian paired *t*-tests were applied to the symmetry and femininity data with default Cauchy priors. For symmetry, the alternative hypothesis is that perfect symmetry is more attractive than normal levels of symmetry. With a $BF_{01} = 0.067$, this represents strong evidence for the null hypothesis, that perfect symmetry does not increase attractiveness ratings, using rules of thumb for the Bayes factor interpretation [90]. Intriguingly, for the femininity comparison, a $BF_{01} = 0.331$ indicated an effect bordering between anecdotal and moderate evidence for the null hypothesis. This suggests that the effect of femininity increasing attractiveness may be present, but the size of the effect is substantially lower than is claimed by previous research [4].

Finally, linear mixed effects models were employed to arrive at estimates of the effects of the factors while accommodating variances due to participants and faces. For each trait, the attractiveness ratings were regressed against the manipulation level (normal or manipulated), with random intercepts for participants and faces. Conceptually, this accounts for baseline differences in the attractiveness of faces and the attractiveness ratings by participants, while assuming the effect of manipulation is identical for all faces and participants. Random slopes were also included for the manipulation across both participants and faces, effectively modelling how the manipulation may affect each face differently, as well as the perceptions of each observer. This model setup is the maximal specification for the given design, as well as the more theoretically important one [91].

For symmetry, the model revealed no evidence of symmetry being more attractive: $b = -0.06$ $[-0.006, 0.130]$, $SE = 0.03$, $t(36.59) = 1.77$, and $p = 0.085$. In fact, the direction of the effect was that the unmanipulated faces had higher ratings.

For averageness, there was a significant effect of the more average faces being perceived as more attractive: $b = 0.39$ $[0.282, 0.497]$, $SE = 0.05$, $t(63.49) = 7.12$, and $p < 0.001$.

For masculinity ratings, the model indicated a negative effect of masculinity, with the unmanipulated faces rated as more attractive: $b = 0.30$ $[0.234, 0.360]$, $SE = 0.03$, $t(46.54) = 9.29$, and $p < 0.001$.

For the femininity ratings, the model indicated no significant difference between the manipulated and unmanipulated versions of femininity, though the direction indicated higher ratings for the manipulated versions of femininity: $b = 0.05$ $[0.028, 0.132]$, $SE = 0.04$, $t(45.87) = 1.27$, and $p = 0.212$.

Manipulation Check—Are Manipulations Too Subtle?

A simple but plausible explanation for the above null results is that the level of the manipulation used were either too subtle to be detected or too extreme and appeared unattractive. Either of these points could explain the results seen in Study One and above. That is, symmetry and femininity may be attractive, but the manipulations are too extreme to capture this. To rule this out, facial femininity was measured using a geometric vector approach (outlined in detail in Study Three) that measures the position of a face on a vector between the average male and average female face. This gave a distribution of the facial femininity in the sample (*M* = −2356.15, *SD* = 59.08). After the

manipulation, the average femininity score was within this range ($M = -2319.55$, $SD = 59.07$). However, the manipulation did shift faces along this vector ($t(108) = 1340.36$, $p < 0.001$). For symmetry, the manipulation itself is consistent with the theory—as the faces are perfectly symmetrical, there is no a priori reason to suspect this should do anything but increase attractiveness.

### 3.3. Discussion

The current study used a conservative methodology to test how attractive the established factors of attractiveness are under more naturalistic settings. By viewing faces that were manipulated or unchanged but never viewing the same stimulus identity more than once, it is possible to make causal claims about the factors and more directly examine their absolute effect sizes.

Standard analyses confirmed that an increased averageness increased attractiveness but an increased masculinity decreased attractiveness, when compared to unmanipulated faces. These effect sizes (averageness $d = 0.88$, masculinity $d = 0.78$) were similar in magnitude to one another, mirroring the magnitudes and directions of the effects established using a forced choice approach. Notably, the effect sizes were smaller compared to the estimates obtained using that methodology. Converging evidence emerged from the use of linear mixed models, which accounted for the variance in the baseline attractiveness of faces and the ratings made by participants and, for the case of averageness, how the manipulation affected face differently. A simple $z$-test comparing the fixed effects from the averageness and masculinity models revealed no difference between their effects on attractiveness ($z = 1.46$, $p = 0.143$). In female faces, averageness increases while masculinity decreases the attractiveness. However, consider the absolute magnitude of the effect size for these factors—relative to an unmanipulated face, the effect for averageness was $b = 0.39$ and for masculinity was $b = -0.30$. Interpreted using the original scale used to collect data, these values amount to less than half a point along this seven-point scale. These effects, despite their robustness, do not influence the absolute ratings of attractiveness considerably.

The analysis for symmetry indicated no effect of the trait on attractiveness perceptions, and the application of Bayesian probabilities indicated a strong support for symmetry not affecting attractiveness. Under more natural conditions, symmetry played almost no role in attractiveness. Indeed, the linear mixed model indicated that, accounting for the baseline differences in attractiveness of the faces and the participant ratings, there is, if anything, a slightly negative effect of perfect symmetry on attractiveness, though this effect is extremely weak. These findings confirm the effects found with forced choice paradigms. Even under that sensitive approach, symmetry still has a small effect, but more conservative tests indicate little evidence of its role in attractiveness. This adds an extra layer to the research showing its evolutionary utility is questionable [14,23].

The evidence for femininity was mixed. The standard univariate analysis indicated that increasing femininity in faces offered no increases in attractiveness, which was supported by linear mixed modelling. The second analysis has the benefit of taking into consideration the baseline attractiveness in female faces, which may be important if a manipulation of femininity operates inconsistently across faces, i.e., some faces appear more attractive, but others appear less. The mixed model approach indicated no evidence of this. However, applying Bayesian tests to the simple univariate analysis indicated borderline evidence between moderate and anecdotal evidence for the null hypothesis. A conservative interpretation of these effects is that an increased femininity in female faces does not lead to increased attractiveness or that the effect is so small it requires a larger sample to detect—directly in opposition to how the effect is discussed in the literature, likely because of methodological weaknesses [3,4].

The findings of the first two studies raise important implications for attractiveness research and face perception more generally. The differences that emerged between studies call into question the equivalence of the designs, as they do not always provide converging evidence—particularly in the case of symmetry. The two studies also question the validity of the two-alternative forced choice method, which is much more common in the literature [3,4] for assessing the effects of facial

attractiveness across a variety of domains. There were clear reasons to investigate that a two-alternative forced choice may produce spurious results, and these have been confirmed across these studies.

## 4. Study Three—Machine Learning Approaches

The above studies indicate that the way attractiveness has been studied in the past has inflated the magnitude of certain effects and skewed the interpretation of others and that even when robust effects are found, their actual contribution is relatively small. From a theoretical perspective, these are not encouraging findings. The study of human attractiveness with an evolutionary approach has borrowed heavily from animal models [16,92] and focused on the explanatory power of these factors in human judgments of attractiveness. The current research and recent findings [13–15] indicate that these effects are not as relevant to attractiveness in an absolute sense as previously thought.

However, the body of work is substantive, and no single study is enough to overturn the established theories. As such, it is useful to consider alternative perspectives in interpreting the utility of the established factors of attractiveness. Researchers have recently argued for the benefits that psychology can gain by bringing to bear the analytical approaches from the field of machine learning (ML) to understanding behavioural phenomena [93]. A key difference between these approaches is that machine learning emphasises predictive accuracy, where the goal of an analysis is to minimise the errors in prediction as much as possible without concern for the reasons behind the minimisation. In contrast, psychology seeks an explanation—finding variables that explain the most variance in a measure of behavior and aiming to conclude that one influences the other. However, these are often conflated in psychological research. For example, some researchers find a very small $R^2$ value for physical measurements of factors like averageness and symmetry, suggesting their relative unimportance [78], while others have found large $R^2$ values using perceptual measures [51] for the main factors of attractiveness [67], emphasising their explanatory power in understanding attractiveness perceptions.

However, the exclusive focus on metrics like $R^2$ in attractiveness research (and psychology broadly) is misleading. A high or low value of $R^2$ offers information on how close a least-squares solution is to each data point and no information about how well a model can actually predict an outcome [94], which alternative metrics like root-mean-square-error (RMSE) can. More problematically, as psychologists rarely test developed models on out-of-sample data through the use of cross validation, the high explanatory power of factors of attractiveness may be due to overfitting—models being too closely aligned to noise in datasets that will fail to generalise. Even when researchers use cross validation, the focus is typically on $R^2$, which does not inform predictive utility [95] or attempt to minimise the importance of theoretical factors by contrasting them with high dimensional models with no correction for the higher number of predictors [96]. More simply, facial attractiveness research has not addressed how well the theoretical factors can, in an absolute sense, predict the perceived beauty of a face in any convincing way.

In this final study, the predictive utility of symmetry, averageness, and sexual dimorphism for facial attractiveness are assessed using novel analytical approaches from the field of machine learning. The three traits are measured directly from faces using geometric techniques common in biology [97] and anthropology [33], which use all available data from faces, rather than being constrained by a select few landmarks or skin regions that is typical in the literature [15,22,98]. These approaches can also negate the limitations the manipulation approach has: For example, symmetrising faces removes both fluctuating and directional asymmetries. A random forest regressor [99] is applied to these predictors, which is capable of a high predictive accuracy as well as capturing the complex nonlinear patterns and interactions in data. These models also provide a useful metric of "feature importance"—a weight similar to a regression beta value that gives the importance of a particular predictor in making decisions about the outcome. This analysis will provide evidence of both the predictive power of the long-standing theoretical approaches in attractiveness research as well the relative importance of the factors in predicting attractiveness that may mesh with the experimental evidence above through the use of "bottom up" approaches in terms of feature measurement and modelling.

If the established factors are important in perceived attractiveness, then a random forest should be able to utilise them to accurately predict the attractiveness of a face, given its symmetry, averageness, and sexual dimorphism.

### 4.1. Method

To apply the random forest, each of the 109 faces used in Studies One and Two were measured for symmetry, averageness, and sexual dimorphism using vector-based geometric techniques, the 169 landmarks placed to manipulate them, and their full RGB colour photographs. The average landmark configuration and facial appearance is shown in Figure 3.

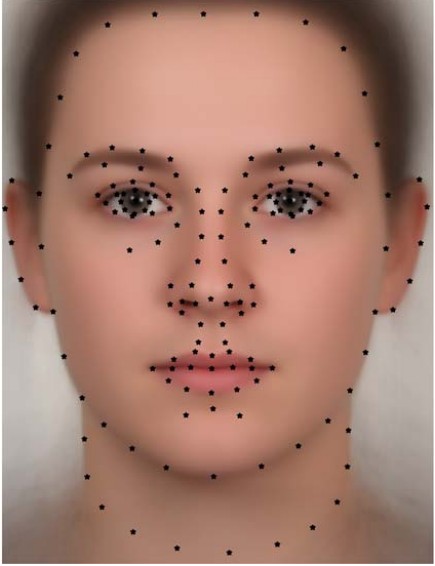

**Figure 3.** The average face of all female and male faces used in the study, representing the average landmark configuration and facial appearance.

**Symmetry**: Rather than compute the summed distances between pairs of bilateral points [8,22], which relies on a small number of landmarks, the full 169 landmark coordinates were treated as a 338-dimensional vector in a Euclidean space (akin to simply flattening the 2-D coordinate array). The coordinates for each face were reflected about the vertical midline, and a difference vector between the original and reflected shapes was computed by subtracting one flattened array from the other. The magnitude of this vector indexed the symmetry of the face, with higher values indicating greater asymmetry. Geometrically, the original and reflected vectors are further away from one another in space, meaning a greater asymmetry—a perfectly symmetrical face would have a vector pointing in the exact same direction with a complete overlap. The same process was repeated on the RGB photograph of each face to assess skin colour symmetry. However, each face was first warped to the average facial shape, and the areas outside of the face were masked. The magnitude of the difference vector between the normal and reflected version was computed as a measure of the texture asymmetry.

**Averageness**: The shape averageness was computed through the following steps. First, the average female shape was computed and flattened to a vector representation. This was repeated for each face, and the difference vector between each face and the average female shape vector was computed. The magnitude of each of these vectors was taken as a measure of averageness—a larger number indicated a shape further from the average. This process was repeated for the vector representation of the RGB images, computing the distance in the shape-normalised texture from the average face.

**Sexual dimorphism**: Sexual dimorphism was assessed using a vector projection technique from anthropology [33]. The technique works through the use of multivariate regression, wherein the

dependent variable is a two-dimensional array of numbers and the independent variables may be one or more variables as in a standard multiple regression. Multivariate regression yields an array of beta-weights, where each column in the dependent variable has a beta value describing its change in a unit of the independent variable. The goal here was to describe the ways in which the biological sex results in changes in both the facial shape and texture. For facial shape, this was achieved by setting the facial landmarks of all female faces and the additional 72 male faces into a 2-D array, with one row per face and columns for each coordinate value. The independent variable was a dummy-coded value representing the sex of each face (1 for female, −1 for male). The least-squares solution, as in a standard multiple regression, was then computed. This yielded a vector of weights that described how each landmark position changes from male to female faces. This vector has the unique property of being "an axis of change" in the dependent variable for the independent variable and can be treated as a Euclidean vector. Carrying out a standard vector projection for each face's landmarks (as a flattened array) onto this axis results in an objective sexual dimorphism score—their position along this derived axis that describes how female and male faces differ. This technique shows good correlations with the actual perceived dimorphism [100] and has been successfully applied to other facial traits like body mass index [101]. The same kind of multivariate regression was carried out for facial texture but simply using the flattened pixel array as the dependent variable—thus, describing the changes in pixel values between images of female and male faces.

The above measures yielded six scores per face, a measure of shape and colour symmetry, averageness, and sexual dimorphism. From the data obtained in Study Two, the ratings of the unmanipulated version of each face were averaged across all participants and used as the ground truth values for applying the model.

Analytic Strategy

Random forests are ensemble machine learning algorithms that build on the power of decision trees, a simple but powerful approach to modelling data that mimics human decision-making and is easily understood [102]. Decision trees split predictors into a series of rectangles, utilising a set of rules that identify regions where the outcome variable is as homogenous as possible. These rectangles can be thought of as simple queries of the data that help make a decision of the outcome, likened to the popular "20 Questions" game [103]. Where the predictors are split (i.e., is facial symmetry above *X*?) is chosen to minimise the prediction errors through least-squares solutions [104]. The trees extend through a series of these splits until a stopping criterion is reached. Predictions can then be made for new data by following the tree's splits down to a terminal node, or leaf, which contains the average response of the outcome variable of all cases that fall within. While decision trees have been adopted in fields such as ecology [105], they suffer from a tendency to overfit—that is, they model noise in the training data, which means they generalise poorly to novel data [104].

The random forest [99] overcomes this inherent weakness of decision trees by building a large number of decision trees, with the goal of obtaining a set of weak or rough predictors that can be averaged over to provide highly accurate predictions [106]. The random component of the random forest algorithm refers to the fact that each individual decision tree is built on a bootstrapped-with-replacement sample of the data and considers a random subset of the available predictor variables—this introduction of randomness ensures no tree is identical and that the issue of overfitting is largely negated [104]. Predictions for new data are made by asking each tree in the forest to make a decision on new data based on what it has learned, and then averaging the predictions across all trees.

Unlike ordinary least squares regression, random forests have a set of hyperparameters that are set before the model is run and that affect its predictive ability by controlling model complexity. The optimal values for these parameters are tuned through cross validation. The most important parameter for a random forest is the number of trees (*nt*), where more trees provide better predictions. There are several parameters that influence the complexity of the individual trees themselves, which

are responsible for minimising the errors in prediction. These are the minimum number of samples required to split a node in a tree (i.e., how many samples should meet the criteria before the node can be split?), the minimum samples per leaf (how many samples should be in each terminal node?), the maximum depth of the tree (how many steps should the tree contain, where more increases the model complexity), and the maximum number of features allowed to be considered by a given tree (lower values reduce overfitting by constraining trees to fewer features). The optimal values for these hyperparameters were derived through a cross-validated grid search, shown in Table 1, using a general range of values indicated in the literature [104,106,107].

**Table 1.** The hyperparameter spaces tuned via a grid search for the random forest.

| Hyperparameter | Grid Search Parameter Space | Nested Cross Validation (Reduced Parameter Space) |
|---|---|---|
| Number of trees (*nt*) | <u>100</u>, 200, 300, 1000 | 80, 100, 120 |
| Maximum features | 3, 4, 5, <u>6</u> | 6 |
| Minimum samples to split | <u>3</u>, 5, 7, 10, 15 | 3, 6 |
| Minimum samples in leaf | 8, <u>10</u>, 12 | 8, 10, 12 |
| Maximum tree depth | 2, <u>3</u>, 5, 10 | 2, 3 |

Note: The optimal values are underlined for the initial grid search, which acted as the basis for the reduced hyperparameter space for the nested cross validation. The model achieved its best grid search root-mean-square-error (RMSE; 0.79) with a relatively small number of trees, the full predictor set, and a shallow tree depth.

### 4.2. Results

In order to arrive at the optimal values for the hyperparameters, the sample of 109 faces was split into a training ($n = 88$) and a validation set ($n = 21$). The training data was used to conduct a five-fold cross-validated grid search across the values for the hyperparameters in Table 1. That is, the training data was split into five folds, and a combination of hyperparameters were used to fit a random forest on four of these folds. This model's predictive accuracy was assessed using RMSE on the fifth fold. Given that a single split of data does not generalise well, the same combination of hyperparameters were fit on another of the four folds, holding out a different fold for assessing accuracy. This was repeated until all folds had been held out once. The average RMSE score was computed across these five folds, returning an accuracy measure for that particular set of hyperparameters. This process was repeated for all combinations of hyperparameters: a total of 960 combinations. The combination of parameters with the lowest RMSE was then used to predict the novel validation data, yielding a RMSE of 0.66—the model predicted facial attractiveness to within 0.66 of a point on the scale, on average, on data it had not encountered before.

However, these hyperparameters were derived from a single split of the data into training and validation sets and do not indicate how well a random forest can do on the existing dataset on average: Different parameter sets would have been derived on different initial splits. Nevertheless, the initial grid search indicated an optimal performance on a particular configuration of valid hyperparameters. Next, a nested five-fold cross validation was carried out, where a reduced set of hyperparameters were tuned on each split based on the initial grid search (see Table 1), using its own five-fold cross validation, before predicting the held-out fold with the tuned model. This process was repeated 20 times (a total of 100 splits). At each iteration, the RMSE and $R^2$ values of the training and test data were carried out, as well as the feature importance of the models. These are highlighted in Feature 3. The feature importances for a single decision tree are calculated by the amount a feature reduces the mean squared error in a node, weighted by the number of observations that reach the node. This is computed for each feature per tree before each features' importance is averaged over all trees in the forest [99].

Across the 100 splits, the model's prediction accuracy on the test data was $M_{\mathrm{RMSE}} = 0.77$, $SD = 0.10$ [0.57, 0.97] and for the train data was $M_{\mathrm{RMSE}} = 0.66$, $SD = 0.04$ [0.59, 0.73], indicating that an optimised random forest was able to predict attractiveness to less than one point on the rating scale, using all

available predictors. However, $R^2$ scores differed greatly between the test data ($M_{R^2}$ = −0.01, $SD$ = 0.17 [−0.35, 0.34]) and train data ($M_{R^2}$ = 0.31, $SD$ = 0.06 [0.20, 0.42]). A relatively high $R^2$ training score versus a relatively negative test score (indicating the model has a worse fit than simply predicting the grand mean attractiveness score for each face) suggests overfitting—the model may be overly complex, and thus, the fitting noise in the training data generalises poorly to new data.

To examine this, a five-fold cross validation was conducted with a simple multiple linear regression, which has no hyperparameters. This was repeated 20 times to match the 100 scores obtained with the cross-validated random forest. The simple regression model had a similar pattern between the test prediction accuracy ($M_{RMSE}$ = 0.80, $SD$ = 0.12 [0.56, 1.04]) and the train prediction accuracy ($M_{RMSE}$ = 0.74, $SD$ = 0.03 [0.68, 0.80]), as well as between test $R^2$ ($M_{R^2}$ = −0.11, $SD$ = 0.27 [−0.64, 0.42]) and train $R^2$ ($M_{R^2}$ = 0.13, $SD$ = 0.04 [0.05, 0.21]). A less complex model still failed to explain the variance in new samples while offering a qualitatively similar predictive power.

One of the benefits of random forests is that they offer feature importances or a measure of what feature the individual decision trees relied on to split the samples. These are highlighted in Figure 4. To examine the important features for predicting attractiveness, a repeated measures ANOVA was conducted on the cross-validated feature importance, examining the influence of Feature Source (Shape and Texture) by Trait (Sexual Dimorphism, Averageness, and Symmetry) on the feature importance scores. The ANOVA revealed some general information about the importance of the factors of attractiveness. There was a main effect of Feature Source ($F(1, 594)$ = 685.12, $p < 0.001$), with Shape ($M$ = 0.24) having a significantly higher importance than Texture ($M$ = 0.10). There was also a main effect of Trait ($F(2, 594)$ = 190.36, $p < 0.001$). This main effect was driven by significant differences between each trait, with Averageness ($M$ = 0.23) being significantly higher than Sexual Dimorphism ($M$ = 0.16) and Dimorphism being higher than Symmetry (0.11); all $ts > 8.97$, all $ps < 0.001$. The interaction was also significant ($F(2, 594)$ = 474.41, $p < 0.001$). This interaction was explored by comparing the three factors (symmetry, averageness, and dimorphism) for shape and texture separately. For shape, averageness ($M$ = 0.38) had significantly higher scores than sexual dimorphism ($M$ = 0.26) and symmetry ($M$ = 0.06); both $ps < 0.001$. Sexual dimorphism had significantly higher scores than symmetry ($p < 0.001$), indicating that, for shape, feature importance increased linearly from symmetry to sexual dimorphism to averageness. For texture, a different pattern emerged, with symmetry ($M$ = 0.15) having significantly greater scores than sexual dimorphism ($M$ = 0.06) and averageness ($M$ = 0.08); $ps < 0.001$. Averageness was only borderline significantly different from sexual dimorphism ($p$ = 0.05).

## 4.3. Discussion

The application of facial geometric measures and a robust machine learning algorithm and analysis procedures revealed converging evidence with the experimental results from Study One and Two, as well as novel information. A tuned random forest relied on different components of appearance to predict attractiveness. More specifically, the most important features were, in descending order, shape averageness, shape dimorphism, and texture symmetry, with the other features contributing relatively little. These data-driven findings support the experimental findings in that averageness and dimorphism stand out as important predictors but with averageness being more important than dimorphism. Notably, shape symmetry was almost entirely unused by the random forest to make decisions on attractiveness. However, texture symmetry, a measure of the evenness of colouration on either side of the face, was informative. Indeed, texture symmetry has been shown to be attractive in a range of research [108–110]. Interestingly, a measure of texture sexual dimorphism was relatively more or less unused, despite aspects of sex differences in skin tone and appearance being rated as attractive [111–114].

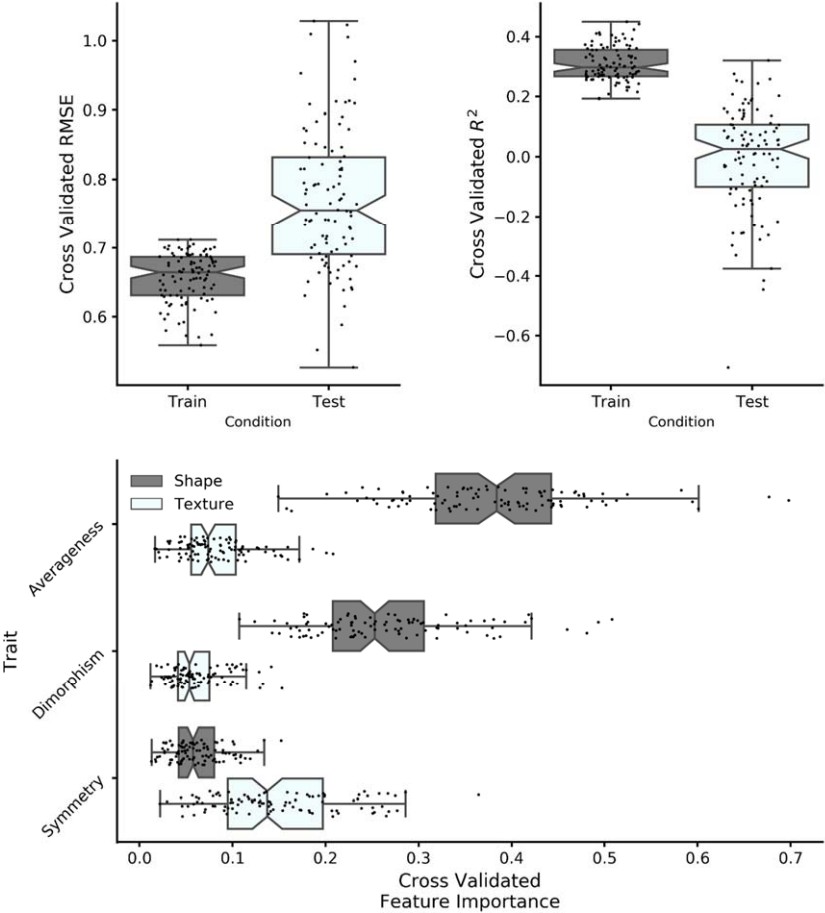

**Figure 4.** The results from the cross-validated random forest on the full dataset: The top left represents the RMSE for training and the test sets for each fold and the top right represents the $R^2$ for the training and test sets. The bottom depicts the feature importance from each fold, highlighting the importance of shape averageness, as well as the utility of texture symmetry. The black dots represent actual data points.

The random forest achieved good accuracy in predicting facial attractiveness on held-out data. In real terms, the mean RMSE score of 0.77 indicated the model could discriminate between faces rated as a six or seven on the scale or as a three or four, for example. In absolute terms, this could be considered a fairly good performance, and the predictions made by the model were similar to human judges. This lends support to the notions that the theoretical factors of beauty are indeed useful metrics to predict how attractive a face is perceived to be. However, what was clear was that the factors of attractiveness themselves generalise poorly in terms of their explanatory power; while they showed respectable training $R^2$ scores, the test $R^2$ scores were very poor and, in most cases, worse than including no predictors at all. This is a clear case where a variable's predictive utility differs from its explanatory utility [93]; facial attractiveness is clearly subsumed by far more factors than what current theories have described, even if those theoretical factors do offer some predictive ability. Other factors like facial adiposity and body mass index [115,116] or representational sparseness [117] may contribute more to the predictive power and to explaining more variance [79].

## 5. General Discussion

Research in the last 30 years has seen symmetry, averageness, and sexual dimorphism linked to a range of putative reproductive benefits through honest signalling [8] and "good genes" hypotheses [39], with these explanations failing to replicate under more thorough investigations [13,14,23]. The current

study employed a range of methods, experimental and statistical, to more clearly define the role of these factors in perceiving facial attractiveness in female faces.

Study One highlighted that the two-alternative forced choice paradigm common in much attractiveness research, while sensitive, inflates the effect sizes and conflates the preferences of sexual dimorphism. Feminised faces were actually not preferred to unmanipulated faces, but rather masculinised faces were avoided. These findings were also robust when taking into account the different appearances of the faces and how a given manipulation affected them.

Study Two demonstrated that, using a conservative presentation paradigm, only averageness and masculinity affected the perceived attractiveness in divergent directions, with average faces being perceived as more attractive and masculinised faces being perceived as less attractive. However, the absolute magnitudes of these effects were small.

Study Three applied geometric face measurements of the factors and machine learning techniques for predicting attractiveness ratings, to complement the manipulation studies. This approach supported the importance of shape averageness and sexual dimorphism, as well as texture symmetry, in perceiving attractiveness. It also demonstrated that while the factors offered good predictive utility, they only explained the variance to the levels seen in the literature [51] on the training but not on the test data, indicating that they do not generalise well or do not explain as much variance in attractiveness as previously thought.

Taken together, these findings indicate that current theoretical factors of beauty significantly influence perceptions of female attractiveness, but the overall effects are weaker than previously thought, and do not explain a large amount of variance in the overall attractiveness perceptions of female faces. Combined with their weak evolutionary utility, the current evidence indicates that the parameters that influence facial attractiveness are mostly unsupported by an overarching theory and have only a small impact on absolute attractiveness perceptions. Indeed, recent work has highlighted the role of observer characteristics in perceiving attractiveness [118], indicating the explanatory factors that comprise a given attractiveness judgment are likely to be much more complex than evolutionary theories have so far predicted. However, the studies here also highlight and confirm the important role of averageness in female facial attractiveness, as well as a caveat with sexual dimorphism: Increased femininity is not necessarily more attractive in female faces, but increased masculinity seems to reliably lead to lower ratings. Conversely, the effect of shape symmetry seems almost entirely unimportant, and its dominance in the literature can possibly be attributed to the methodologies used to assess it. However, texture symmetry is somewhat important, and this is borne out by dermatological studies that show smooth, even skin texture is attractive [119,120].

There are two limitations worth considering when interpreting the evidence here. The first is that the faces used are Caucasian, which limits the generalizability of the findings. In fact, recent evidence indicates that preferences for the important factors studied here vary geographically, with preferences for femininity being absent in some places but stronger in others [121]. The same study showed that African populations place a higher value on skin colouration than aspects of facial shape, which may speak to the different signalling qualities that skin and shape have in different contexts [15,121]. This means that the small effects observed here may be a consequence of the observers and that these may alter depending on the population studied. The second limitation is that the studies utilised only full-frontal facial photographs. This single viewpoint has been demonstrated as not being able to provide sufficient information about other averageness and sexual dimorphism in other views [122], which would be assessed in a real life interaction. As such, the magnitudes of the effects may well change with exposure to other views of faces.

These findings have important implications for researchers examining attractiveness or the effects of manipulations in face perception research more generally. First, the use of a two-alternative forced choice methodology has the following drawbacks: the relative comparison it provides may skew the theoretical interpretations of the direction of an effect, it may inflate the size of an effect, and it may also produce effects that do not generalise in other methodologies. Second, the sizes of the effects

in attractiveness research are generally small, and in some cases (such as averageness), there is a significant variation in how stimulus items are affected by manipulations. Researchers should consider the use of more sensitive statistical designs to better isolate these effects. Finally, the results from Study Three indicate that while the theoretical factors do offer some predictive power about attractiveness, their absolute explanatory power is poor. Future research should consider the benefits of a data-driven approach to studying facial attractiveness [123]. Existing ventures in these areas have yielded a range of novel components that affect attractiveness that currently have little theoretical grounding [96] but could nonetheless contribute significantly to the understanding of perceived beauty. Despite the success of evolutionary psychological approaches in understanding human mating, it is perhaps time to significantly broaden the scope of understanding beyond honest signalling approaches to different viewpoints, such as imprinting [124] or Fisherian runaway [125]. Recent evidence has indicated that mate choice decisions are consistent, that people seem to have a "type" of partner they prefer [126,127], and that optimising their preferred traits are what drives attractiveness perceptions. Focusing more closely on these approaches allows for a systematic exploration of the theoretical space of psychological and morphological traits of possible partners.

**Supplementary Materials:** For all experiments reported here, the full data and Python code used to analyse it are available from the Open Science Framework (https://osf.io/uyzmj/) in Jupyter Notebooks.

**Author Contributions:** A.L.J. gathered data, conducted analysis of Study One and Three, and drafted the manuscript. B.J. conducted analysis of Study Two and contributed to redrafts and editing of the manuscript.

**Funding:** This research received no external funding.

**Conflicts of Interest:** The authors declare no conflict of interest.

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
