# Peer review of "Biological Bases of Beauty Revisited: The Effect of Symmetry, Averageness, and Sexual Dimorphism on Female Facial Attractiveness"

_symmetry, doi:10.3390/sym11020279_

Reviewer 1 Report

The manuscript is technically and methodologically sound, and raises important implications for the study female facial attractiveness. The numbers of participants engaged in the study 1 and 2 are sufficient to secure reliable results. I am not able to fully evaluate machine learning approach used in study 3 but altogether its applications (and reported results) are innovative, persuasive, and add importantly to study 1 and 2. I especially appreciate the critical focus of this study towards mainstream theorizing on attractiveness (and associated research heuristics) that apparently needs to be rethought and revised. I have only several minor comments listed below:

 Line 9: sexual dimorphism: I have no problem with current usage of the term but biologists usually consider sexual dimorphism for a “state” which is present or absent and not for “a degree of development” of sexually dimorphic traits. In case of this manuscript we can talk rather about the degree of sexual dimorphism: maleness/femaleness (as termed, for instance, in Mitteroecker et al 2015, Plos One)

Line 14: “femininity, and masculinity,” Shouldn’t we talk rather about maleness and femaleness? If sexual dimorphism and not perceived masculinity/femininity is discussed?

Line 30: “folk wisdom”: I would rather call this a classical notion of beauty in European intellectual tradition, which was also held by Hume, later Darwin, and many others. See for example:

David Hume: „[beauty] is no quality in things themselves: it exists merely in the mind which conteplates them; and each mind contemplates a different beauty (19, pp.208-209).

Charles Darwin: „It is certainly not true that there is in the mind of man any universal standards of beauty with respect to human body“

 Line 152 “Rating studies:”

I think that the rating studies based on natural-non manipulated photos, videos, and 3D scans is the additional option to those authors are presenting in their paper. This could be mentioned for the sake of clarity but I do not insist indeed.

 Line 237: “averageness should have a larger effect than symmetry“: You mean total bilateral symmetry? Directional plus fluctuating asymmetry? Or, some kind of overall symmetry presented by the effect of symmetrized stimuli? 

Line 258: "To manipulate symmetry in faces, the position of corresponding points on the left and right sides of the face, relative to the midline, were averaged"

Do you mean that the paired landmarks were mirrored and relabelled and these mirrored versions then averaged?

Line 270: "To manipulate femininity, 50% of the difference between the male average and female average face was added to each individual face. " I am not sure if understand this. Did you added 50% of the difference between averages to each female face, i.e. those 50% between global mean and female mean?

"To manipulate masculinity, 50% of the difference was subtracted, ..."

The 50% of the difference between global mean and female mean was subtracted from each female face?

Line 561: “The three traits are measured directly from faces using geometric techniques common in 561 biology [94],…” Why you talk about biology while referring to (anthropological?) Mitteroecker’s et al 2015 paper about morphological measurements of masculinity/maleness from human facial photographs. Maybe authors can refer to some more general methodological introductory paper on GM methods by James Rohlf, Fred Bookstein or so.

Line 580: The image of landmark position of a face would be helpful

Line 584: "Geometrically, the original and reflected vectors are further away from one ..." Is this sentence necessary? I am not sure what kind of information it has for a reader.

Line 587: "and the non-face regions of the image were masked"

Did you masked only areas out of the face, or did you exclude also eyes?

Line 790: besides imprinting and Fisherian runaway, one can consider also recently evidenced “consistency” in mate choice, and/ or selection of partners based on the inherent possession of “a type” that helps to systematically explore the theoretical space of phenotypic/psychical traits of possible mating partners.

 Finally, I have two comments to “general discussion” based on our own recent research.

 1) I think that if we focus at the preferences in some non-European cultures the results will be even more ambiguous. In our African rating study (Evolution and Human Behavior 38 (6), 744-755), we found none or only weak preferences for feminine traits (sex. dimorphism) and averageness using non-manipulated faces. Nevertheless, skin colour was a decisive cue to attractiveness for African male perceivers.

 2) One can also speculate that frontal views of the faces do not provide enough information for assessment of averageness and degree of sexual dimorphism because frontal and profile facial configurations are only weakly/moderately correlated. See: Danel et al 2018. A cross‐cultural study of sex‐typicality and averageness: Correlation between frontal and lateral measures of human faces American Journal of Human Biology 30 (5), e23147.

 I enjoyed reading your paper,

Karel Kleisner

Author Response

Reviewer 1 comments

 The manuscript is technically and methodologically sound, and raises important implications for the study female facial attractiveness. The numbers of participants engaged in the study 1 and 2 are sufficient to secure reliable results. I am not able to fully evaluate machine learning approach used in study 3 but altogether its applications (and reported results) are innovative, persuasive, and add importantly to study 1 and 2. I especially appreciate the critical focus of this study towards mainstream theorizing on attractiveness (and associated research heuristics) that apparently needs to be rethought and revised. I have only several minor comments listed below:

 Point 1: Line 9: sexual dimorphism: I have no problem with current usage of the term but biologists usually consider sexual dimorphism for a “state” which is present or absent and not for “a degree of development” of sexually dimorphic traits. In case of this manuscript we can talk rather about the degree of sexual dimorphism: maleness/femaleness (as termed, for instance, in Mitteroecker et al 2015, Plos One)

 Point 2: Line 14: “femininity, and masculinity,” Shouldn’t we talk rather about maleness and femaleness? If sexual dimorphism and not perceived masculinity/femininity is discussed?

Response 1 & 2: These are both good points, and we are inclined to agree with the reviewer that the more biological and accurate description would be maleness/femaleness. However, almost the entire literature base uses the terms of femininity and masculinity when referring to sexual dimorphism, and since these are the approaches and methodologies we wish to critique in this manuscript, we would like to retain the current terminology. However, as we agree with these points, we have added an explanation of this terminology to the sexual dimorphism section in the introduction. We hope that this is a satisfactory outcome for the reviewer.

 Point 3: Line 30: “folk wisdom”: I would rather call this a classical notion of beauty in European intellectual tradition, which was also held by Hume, later Darwin, and many others. See for example:

David Hume: „[beauty] is no quality in things themselves: it exists merely in the mind which conteplates them; and each mind contemplates a different beauty (19, pp.208-209).

Charles Darwin: „It is certainly not true that there is in the mind of man any universal standards of beauty with respect to human body“

Response 3: This is true. The opening paragraph has now been amended to include mention of this classical approach to better frame the introduction.

Point 4: Line 152 “Rating studies:”

I think that the rating studies based on natural-non manipulated photos, videos, and 3D scans is the additional option to those authors are presenting in their paper. This could be mentioned for the sake of clarity but I do not insist indeed.

Response 4: While the focus is on manipulation studies in this section, it is true that the range of natural, non-manipulated face studies with different stimulus types support these notions. Near line 152 a sentence has been added with some extra citations to reflect this.

Point 5: Line 237: “averageness should have a larger effect than symmetry“: You mean total bilateral symmetry? Directional plus fluctuating asymmetry? Or, some kind of overall symmetry presented by the effect of symmetrized stimuli?  

Response 5: Apologies for not making this clear. The manipulation will improve total bilateral symmetry, so correcting directional and fluctuating asymmetry in faces. This has now been made clearer. This is of course a methodological weakness, as humans may be sensitive to both FA and DA, and changing both at once may lead to erroneous conclusions – this has been added to the motivation of Study Three.

Point 6: Line 258: "To manipulate symmetry in faces, the position of corresponding points on the left and right sides of the face, relative to the midline, were averaged" 

Do you mean that the paired landmarks were mirrored and relabelled and these mirrored versions then averaged?

Response 6: That is indeed the approach that was taken. The explanation was taken from the software used to manipulate the faces (JPsychomorph), but your explanation was also added for the benefit of other readers.

Point 7: Line 270: "To manipulate femininity, 50% of the difference between the male average and female average face was added to each individual face. " I am not sure if understand this. Did you added 50% of the difference between averages to each female face, i.e. those 50% between global mean and female mean?

"To manipulate masculinity, 50% of the difference was subtracted, ..."

The 50% of the difference between global mean and female mean was subtracted from each female face?

Response 7: Apologies that this was unclear – this section has been rewritten, and is now hopefully clearer. To be explicit, a female and male average face was created. The linear difference between these averages represented a sexual dimorphism continuum. To each individual face a +50% scaling of this linear difference was added to increase femininity, and a -50% scaling of the linear difference was added to increase masculinity. This is the standard approach that is common in many papers examining the forced choice methodology, typically from researchers in the United Kingdom where it originated, and so it was used here to be in line with those methods that were critiqued.

 Point 8: Line 561: “The three traits are measured directly from faces using geometric techniques common in 561 biology [94],…” Why you talk about biology while referring to (anthropological?) Mitteroecker’s et al 2015 paper about morphological measurements of masculinity/maleness from human facial photographs. Maybe authors can refer to some more general methodological introductory paper on GM methods by James Rohlf, Fred Bookstein or so. 

Response 8: This is a fair point, and there is no intention to mischaracterise the human anthropological work by Mitteroecker et al. This citation has been replaced with one to Bookstein’s seminal book on topic (1992).

Point 9: Line 580: The image of landmark position of a face would be helpful 

Response 9: Absolutely, this has been added as Figure 3 at the start of the Methods section of Study Three.

Point 10: Line 584: "Geometrically, the original and reflected vectors are further away from one ..." Is this sentence necessary? I am not sure what kind of information it has for a reader.

Response 10: This sentence is borne from the first authors grappling with conceptual understanding of faces-as-vectors. It has been expanded to be informative to readers who may be unfamiliar with these vector techniques. It has now been made clear that, for example, a perfectly symmetrical face would have a reflected vector indistinguishable from its original, and so vectors further away indicate greater asymmetry.

Point 11: Line 587: "and the non-face regions of the image were masked" 

Did you masked only areas out of the face, or did you exclude also eyes?

Response 11: Only areas outside of the face were masked, which has now been made clearer. This is more important from the skin texture point of view – i.e. one reddened sclera might be indicative of disease and so on.

Point 12: Line 790: besides imprinting and Fisherian runaway, one can consider also recently evidenced “consistency” in mate choice, and/ or selection of partners based on the inherent possession of “a type” that helps to systematically explore the theoretical space of phenotypic/psychical traits of possible mating partners.

Response 12: Thank you for this suggestion. The conclusion of the manuscript has been altered to focus more closely on this, and has included citation of two recent manuscripts in the area.

Point 13: Finally, I have two comments to “general discussion” based on our own recent research.

1) I think that if we focus at the preferences in some non-European cultures the results will be even more ambiguous. In our African rating study (Evolution and Human Behavior 38 (6), 744-755), we found none or only weak preferences for feminine traits (sex. dimorphism) and averageness using non-manipulated faces. Nevertheless, skin colour was a decisive cue to attractiveness for African male perceivers.

2) One can also speculate that frontal views of the faces do not provide enough information for assessment of averageness and degree of sexual dimorphism because frontal and profile facial configurations are only weakly/moderately correlated. See: Danel et al 2018. A cross‐cultural study of sex‐typicality and averageness: Correlation between frontal and lateral measures of human faces American Journal of Human Biology 30 (5), e23147.

Response 13: These are both important limitations that frame the interpretation of the data presented in the manuscript. An extra paragraph has been added to the discussion to draw attention to these – thank you for pointing them out.

I enjoyed reading your paper, 

Karel Kleisner

Thank you for your helpful comments, and we are glad you enjoyed the paper!

Reviewer 2 Report

This is a very well-written manuscript that combines 3 different methodologies in an interesting way to compare the effects of averageness, symmetry and dimorphism on female facial attractiveness. I think the paper would make an excellent contribution to the literature on factors leading to attractiveness. However, there is a lot of revision that is needed first, because the paper reads too much like a review paper, with methods and results/conclusions added in (or in some cases not added in) without much explanation.

Abstract: 

The abstract is incomplete and a bit misleading. For experiment 1, the effect of symmetry is not mentioned and the effect of femininity presented in a confusing manner. The description of experiment 3 doesn't mention the interesting effect of texture symmetry. Even if other factors were more important, symmetry should at least be mentioned: as described it seems like symmetry had no effect at all.

In the first sentence, "theoretical factors" sounds odd. I think you could just delete "theoretical".

Introduction:

The introduction is well-written, but probably too long. A lot of what is written here could probably be removed or condensed without losing much.

lines 40-43. I know what the authors are saying, but this part is worded in a confusing manner.

lines 49-51. FA can also indicate a lack of genetic defenses vs. these "insults". Also, replace the authors' names/years with numbers in the citation.

Study One:

Much more detail is needed in the Methods:

Under "Participants", differences in the number of individuals is not mentioned (e.g. Why 35 vs. 31 vs. 32 vs. 30 participants), and more importantly, why 12 vs. 11 vs. 8 vs. 11 stimulus females?  Also, it took me some time to figure out that "(12 females....)" represented the stimulus, because it is presented in the same sentence as the number of participants in the study.

Under "Stimuli" it is hard to know what the authors actually did to manipulate the images.

Results: 

Figure 1. Please double check that these are 95% confidence intervals. They seem to be very small.

What is stated on lines 296-298 for femininity seems to contradict what is stated in the Abstract.

Discussion:

First paragraph. There are several meta-analyses that could be cited on the effects of symmetry, so what is stated about the lack of attention to the magnitude of the effects is misleading (e.g. ref. 3, Graham and Özener, 2016, Symmetry, 8, 154; Grebe et al., 2017, Symmetry, 9, 98; van Dongen and Gangestad, 2011, Evol. Human Behav. 32: 380-398). At least some of these and earlier meta-analyses also explain why we expect low effect sizes for symmetry even when symmetry plays an important role (that it is an indirect measure of developmental stability, which in itself is an indirect measure of quality, etc.). The last sentence in this paragraph is also confusing.

Study Two

Participants:  As above, there needs to be explanation for variation in the numbers of those tested and those used as stimuli. This could be written just in Study One, assuming the reasons are the same.  It is also probably fine to do this, but it seems strange that subjects came from different locations (North American college vs. UK university), and this difference is not commented on.

Results: Usually a power level of 0.8 is used, rather than 0.9. Any reason for the higher power?

Study Three

Lines 540-542. The difference between “prediction” and “explanations” needs to be explained.

Methods

Again more detail is needed, especially under “Sexual dimorphism”

The first sentence under “Averageness” (lines 589-591) is confusing as written.

General Discussion

Line 763. It is not clear what “both” is referring to. Does this mean dimorphism and symmetry?

Lines 770-773. Make it more clear that you are referring specifically to attractiveness of female faces.

Lines 774-775. This sentence seems to be contradicted by study 3, where texture symmetry was important.

The last sentence is probably worded too strongly, as the authors’ results show important effects of some of the same traits used in previous studies, just with weaker effects.

Author Response

Reviewer 2 comments

This is a very well-written manuscript that combines 3 different methodologies in an interesting way to compare the effects of averageness, symmetry and dimorphism on female facial attractiveness. I think the paper would make an excellent contribution to the literature on factors leading to attractiveness. However, there is a lot of revision that is needed first, because the paper reads too much like a review paper, with methods and results/conclusions added in (or in some cases not added in) without much explanation.

Abstract:  

Point 1: The abstract is incomplete and a bit misleading. For experiment 1, the effect of symmetry is not mentioned and the effect of femininity presented in a confusing manner. The description of experiment 3 doesn't mention the interesting effect of texture symmetry. Even if other factors were more important, symmetry should at least be mentioned: as described it seems like symmetry had no effect at all.

Point 2: In the first sentence, "theoretical factors" sounds odd. I think you could just delete "theoretical".

Response 1 and 2: Thank you for pointing this out. On reflection, the abstract is substandard and has been completely rewritten. We hope the new version is clearer. 

Introduction:

Point 3: The introduction is well-written, but probably too long. A lot of what is written here could probably be removed or condensed without losing much. 

Response 3: We appreciate that the introduction is long. We have removed several sentences from the paragraphs detailing the backgrounds to symmetry, averageness, and sexual dimorphism in an attempt to shorten the manuscript somewhat. However, we are reluctant to cut too much, given that the manuscript focuses on three traits and two methodologies that are prevalent in the literature, and that the critiques of these are nuanced. Hopefully, the level of editing is satisfactory, but further revisions can be considered if this is not the case.

Point 4: lines 40-43. I know what the authors are saying, but this part is worded in a confusing manner.

Response 4: This has now been reworded, and we hope that it is clearer.

Point 5: lines 49-51. FA can also indicate a lack of genetic defenses vs. these "insults". Also, replace the authors' names/years with numbers in the citation.

Response 5: Thank you for spotting this. It has now been corrected.

Study One:

Much more detail is needed in the Methods:

Point 6: Under "Participants", differences in the number of individuals is not mentioned (e.g. Why 35 vs. 31 vs. 32 vs. 30 participants), and more importantly, why 12 vs. 11 vs. 8 vs. 11 stimulus females?  Also, it took me some time to figure out that "(12 females....)" represented the stimulus, because it is presented in the same sentence as the number of participants in the study. 

Response 6: We apologise that this was not clear, and have amended this paragraph to make it more explicit. The reason for differing participant numbers across the four tasks is simply due to recruitment timelines – the studies were open across a semester and different numbers of people took part in them. The numbers in parentheses actually describe the number of female participants in the sample that completed the tasks, and not the number of stimuli (109 faces) that is described in the following section and is consistent across all four of the studies. While the number of participants does differ between studies, this is not too problematic for the analysis as the data is at first analysed by averaging across participants to provide a score for each face, and secondly is analysed via a general linear mixed model which takes into account variations in participants and should be relatively unaffected by the slight differences in sample size.

Point 7: Under "Stimuli" it is hard to know what the authors actually did to manipulate the images. 

Response 7: This section has also been completely rewritten to address concerns of both reviewers – we hope that the methods are clearer now. The general approach is to compute linear differences between the shapes of faces, and then add or subtract 50% of this difference to individual faces. Practically, this is all handled by the JPsychomorph software that is frequently used in the literature for these kinds of alterations to faces.

Results:  

Point 8: Figure 1. Please double check that these are 95% confidence intervals. They seem to be very small.

Response 8: This has been double-checked and they do represent the 95% confidence intervals.

Point 9: What is stated on lines 296-298 for femininity seems to contradict what is stated in the Abstract.

Response 9: The new abstract has hopefully made this clearer – increased femininity resulted in a preference for the unmanipulated face.

Discussion:

Point 10: First paragraph. There are several meta-analyses that could be cited on the effects of symmetry, so what is stated about the lack of attention to the magnitude of the effects is misleading (e.g. ref. 3, Graham and Özener, 2016, Symmetry, 8, 154; Grebe et al., 2017, Symmetry9, 98; van Dongen and Gangestad, 2011, Evol. Human Behav. 32: 380-398). At least some of these and earlier meta-analyses also explain why we expect low effect sizes for symmetry even when symmetry plays an important role (that it is an indirect measure of developmental stability, which in itself is an indirect measure of quality, etc.). The last sentence in this paragraph is also confusing.

Response 10: Thank you for drawing our attention to these meta-analyses, which are useful and relevant. They have now been added and the implications have been discussed, in that we should expect small effect sizes. In addition, the final line of the paragraph has been redressed – our aim was to make clear that if an experiment has a small number of faces as stimuli, an effect could be driven by a subset of those faces that experience a large change due to a manipulation.

Study Two

Point 11: Participants:  As above, there needs to be explanation for variation in the numbers of those tested and those used as stimuli. This could be written just in Study One, assuming the reasons are the same.  It is also probably fine to do this, but it seems strange that subjects came from different locations (North American college vs. UK university), and this difference is not commented on.

Response 11: This has been made explicit in the ‘Participants’ section. The reason is the same, in that testing was conducted over the course of a semester and different numbers of participants were successfully recruited. The location change was due to the first author changing institutions as part of career progression. Given both are westernised countries, it is unlikely this would affect the results, though this is also commented on.

Point 12: Results: Usually a power level of 0.8 is used, rather than 0.9. Any reason for the higher power?

Response 12: The reason for the higher level of power is to lend additional weight to the interpretation of null effects. We wanted to be more certain that if there was no effect we could conclude this was not a type II error. We also bolstered this with the use of Bayesian approaches to interpret more clearly the non-significant results.

Study Three

Point 13: Lines 540-542. The difference between “prediction” and “explanations” needs to be explained.

Response 13: This has been expanded on in more detail. More simply, machine learning approaches try to reduce the prediction error to be as small as possible without care for the reasons behind this reduction (a variable might be tangential to the problem, but as long as it reduces prediction, it is ‘useful’), while psychology is trying to find a variable that explains the variance in an outcome variable as much as possible. However, there are techniques in machine learning such as cross validation that are essential to overcome the shortcomings of the explanatory approaches psychology takes, which is expanded on further in the following paragraphs.

Methods

Point 14: Again more detail is needed, especially under “Sexual dimorphism”

Point 15: The first sentence under “Averageness” (lines 589-591) is confusing as written.

Response 14 and 15: Thank you for pointing this out. Certainly, this approach is confusing and not enough detail was included here. This section has been almost entirely rewritten with a particular focus on the derivation of sexual dimorphism scores. Hopefully this is now much clearer.

General Discussion

Point 16: Line 763. It is not clear what “both” is referring to. Does this mean dimorphism and symmetry?

Response 16: This was referring to the conclusions that the effects were weaker, and that they do not explain much variance – ‘both’ in this context is misleading, and so has been removed.

Point 17: Lines 770-773. Make it more clear that you are referring specifically to attractiveness of female faces.

Response 17: It has now been explicitly mentioned that these findings refer to female facial attractiveness.

Point 18: Lines 774-775. This sentence seems to be contradicted by study 3, where texture symmetry was important.

Response 18: We agree and this has been changed, with citations to studies showing even skin texture is an important trait for attractiveness.

Point 19: The last sentence is probably worded too strongly, as the authors’ results show important effects of some of the same traits used in previous studies, just with weaker effects.

Response 19: The final sentence has now been altered at the suggestion of both reviewers. It is hopefully clear that there are likely more factors that explain attractiveness that are yet to be unearthed or examined in detail.

Finally, thank you for your review of the manuscript. We hope we have addressed your concerns satisfactorily and look forward to further comments.

Round  2

Reviewer 2 Report

Thank you for addressing the concerns that the other reviewer and I had. You have done an excellent job making these changes in a manner that explains the methodology and conclusions much more clearly.

I have the following very minor corrections/suggestions:

The Abstract is much improved. I see just one small wording problem: In the 1st and 2nd sentence, "studied" and "studies" refers to past studies and then in the 3rd sentence, "studies" refers to your own research. In the 3rd sentence, I would replace "studies use" with "research uses" or something similar.

Study 1, line 270. "measure" should be "measuring"

Study 3, lines 642-643. It sounds odd to refer to BMI as a facial trait. I think you should just delete "facial" before "facial traits".

Last lines of General Discussion. I could be wrong, but I think when the other reviewer wrote "one can consider also recently evidenced “consistency” in mate choice, and/ or selection of partners based on the inherent possession of “a type”" that two different topics were being referred to: 1) something like "mate-choice copying" across individuals and 2) consistency within individuals in the type preferred. The authors only comment on (2). I don't know whether there are human studies on mate-choice copying, but wouldn't be surprised, given the large number of animal studies that address this topic.